# Application of the double bounded dichotomous choice model to the estimation of parent's willingness to pay for the hand foot mouth disease vaccination: A survey in Selangor, Malaysia

Yogambigai Rajamoorthy[1]*, Niazlin Mohd Taib[2‡], Harapan Harapan[3,4,5‡], Abram Luther Wagner[6], Subramaniam Munusamy[7‡]

1 Department of Economics, Faculty of Accountancy and Management, Universiti Tunku Abdul Rahman, Sungai Long Campus, Kajang, Selangor, Malaysia, 2 Department of Medical Microbiology and Parasitology, Faculty of Medicine and Health Science, Universiti Putra Malaysia, UPM, Serdang, Selangor, Malaysia, 3 Department of Microbiology, School of Medicine, Universitas Syiah Kuala, Banda Aceh, Indonesia, 4 Medical Research Unit, School of Medicine, Universitas Syiah Kuala, Banda Aceh, Indonesia, 5 Tropical Disease Centre, School of Medicine, Universitas Syiah Kuala, Banda Aceh, Indonesia, 6 Department of Epidemiology, University of Michigan, Ann Arbor, Michigan, United States of America, 7 Faculty of School of Management and Business, Manipal International University, Nilai, Negeri Sembilan, Malaysia

☯ These authors contributed equally to this work.
‡ NMT, HH and SM also contributed equally to this work.
* yogambigai@utar.edu.my

**Data Availability Statement:** The data underlying the results presented in the study are available from https://figshare.com/articles/dataset/

## Abstract

Hand foot and mouth disease (HFMD) is a notifiable viral disease in Malaysia, and is transmitted primarily among young children. Although vaccines for enteroviruses 71 (EV-71) were approved in China against HFMD, the availability and the acceptance of the vaccine in the Malaysia are unknown. This study investigated and ascertained the determinants of willingness-to-pay (WTP) for HFMD vaccination in Selangor Malaysia. This study adopted a cross-sectional, contingent valuation method involving 390 parents of young children aged six and below. The double bounded dichotomous choice (DBDC) approach was employed to assess the WTP for HFMD vaccine among respondents. A bivariate probit model was used to assess the key determinants of WTP for HFMD vaccine, while the mean WTP was measured using the Krinsky and Robb procedure. We found that 279 (71.5%) of parents were willing to pay for the HFMD vaccination. The estimated single bounded mean WTP was MYR460.23 (equivalent to US$ 102.17) for two doses of HFMD vaccination. The double bounded analysis revealed that the vaccine's price, poor education background and lower income were the key factors that significantly affected the WTP, with the estimated mean WTP being MYR394.00 (US$ 87.47). In conclusion, most Malaysian parents are willing to pay for the HFMD vaccination. The estimated WTP identifies the optimal price point for HFMD vaccination in Malaysia. Furthermore, the government should focus on an awareness programme for the HFMD vaccination among parents who have lower income or education level.

Willingness_to_pay_for_hand_foot_mouth_
disease_vaccination_in_Malaysia/21655349.

**Funding:** This study was funded by Universiti
Tunku Abdul Rahman Research Fund (UTARRF),
grant number IPSR/RMC/UTARRF/2019-C1/Y03.
The funders had no role in study design, data
collection and analysis, decision to publish, or
preparation of the manuscript The funders had no
role in study design, data collection and analysis,
decision to publish, or preparation of the
manuscript.

**Competing interests:** The authors have declared
that no competing interests exist.

## Introduction

Hand foot and mouth disease (HFMD) is a disease caused by several viruses within the *Picornaviridae* family, notably enterovirus 71 (EV71) and coxsackievirus A16 [1, 2]. HFMD viruses are transmissible from person to person by direct contact with nose and throat discharges, saliva, fluid from blisters, or from the stool of infected persons. HFMD is particularly prominent in children <5 years, causing severe blister-like rashes on the hand, foot, and mouth and in rare cases can results in encephalitis, polio-like syndrome and other diseases of the central nervous system [2].

HFMD occurs worldwide, and its global emergence necessitates enhanced surveillance of diseases [3, 4]. Within Malaysia, the outbreaks of HFMD cases are typically due to EV71[3, 5]. The first epidemic of HFMD in Malaysia was in 1997 followed by a subsequent outbreak in late 2000, in 2003, 2005, 2006, 2010 and 2018. Each caused multiple deaths amongst young children [3]. Young children, especially in crowded areas such as in childcare centres, have a high-risk encountering and spreading the disease [6]. It is also possible for more virulent strains of the virus–in terms of neurological complications and mortality—to emerge in the future [3]. Beyond the direct health effects of an EV71 infection, HFMD outbreaks could result in larger socioeconomics consequences [7]. Caring for children with HFMD results in parental absenteeism from work [8]. Moreover, large HFMD outbreaks could result in closure of childcare centres compounding these issues [8].

Recent development of an EV71 vaccine has emerged as an important tool to control HFMD [4, 9]. Treatment of HFMD is largely supportive with no specific therapies available [2]. Since 2015, three inactivated EV-71 vaccines have been approved in China [7] and effectively reduce the severe HFMD cases and mortality caused by EV71 [10]. However, the vaccines have not yet been approved in Malaysia leaving the country with no effective prevention or treatment methods for HFMD [11].

In the future, the EV71 vaccine may be approved for use within Malaysia. However, it is yet unclear how well accepted the vaccine will be by the general population. Presumably, parental willingness to accept the EV-71 vaccines will be affected by its efficacy, side effects, and cost. There are only two studies published in China on the willingness of parents to vaccinate [5, 7]. However, the willingness to pay (WTP) for HFMD vaccination has yet to be estimated, and could also differ between countries based on their epidemiological background and population experiences with disease. Therefore, this study's main objective was to estimate WTP of parents for HFMD and identify its determinants by employing the contingent valuation method in Selangor state.

## Materials and methods

### Ethics approval

The Scientific and Ethical Review Committee of Universiti Tunku Abdul Rahman approved the study protocol (approval U/SERC/17/2020). A brief explanation of the study was given to all participants and written informed concern was obtained from all participants.

### Study design and setting

A cross-sectional survey was conducted in Selangor state, which encircles the capital Kuala Lumpur. Selangor consists of nine districts such as Gombak, Klang, Kuala Langat, Kuala Selangor, Petaling, Sebak Bernam, Sepang, Ulu Langat and Ulu Selangor (Table 1). To represent the population, several kindergartens from each district were randomly selected. The

**Table 1. Sample distribution.**

| No | Districts | Population ('000) | Proportion | Sample size | Double the sample size |
|---|---|---|---|---|---|
| 1 | Sabak Bernam | 126.1 | 0.02 | 8 | 16 |
| 2 | Kuala Selangor | 251.2 | 0.04 | 15 | 30 |
| 3 | Klang | 1025.1 | 0.16 | 60 | 120 |
| 4 | Kuala Langat | 270.1 | 0.04 | 16 | 32 |
| 5 | Petaling | 2157 | 0.33 | 127 | 254 |
| 6 | Sepang | 256.9 | 0.04 | 15 | 30 |
| 7 | Ulu Langat | 1370.2 | 0.21 | 81 | 162 |
| 8 | Gombak | 815.2 | 0.13 | 48 | 96 |
| 9 | Ulu Selangor | 237.6 | 0.04 | 14 | 28 |
| | Total | 6509.4 | 1.00 | 384 | 768 |

sample size and respondents' selection criteria were a part of a Malaysian HFMD Project that have been described elsewhere [12, 13].

The Krejcie and Morgan sampling method was adapted to determine the study sample size [14].

The formula used to calculate the sample size for this study as follows:

$$s = \frac{[z^2 p(1-p)/e^2]}{1 + \{z^2 p(1-p)/e^2 N\}}$$

Where:

 $s$ = required sample size
 $p$ = sample proportion
 $e$ = margin of error
 $N$ = population size
 $z$ = Z-score

As a minimal sample size, 384 participants were required on the following assumptions: (a) 5% margin of error; and (b) 95% confidence level; (c) 50% sample proportion; (d) 1.96 z-score; (e) 6,509,400 population size. The participants were selected using a three-stage cluster sampling method. Using the nine administrative districts in Selangor as a sampling frame, the number of samples from each district was calculated based on its population size proportion (i.e. high numbers in some districts and low in some districts). In the second stage, a convenience sample of kindergartens was selected based on kindergarten lists in each district. In the last stage, a convenience sample of participants was recruited from each kindergarten using a quota to meet each district's calculated sample size. Before starting the study, we doubled the sample size to 768 (≈770) to avoid insufficient sample size due to incomplete data.

The English version questionnaire was developed based on the existing literature and translated to Malay language. The questionnaire used in this study is given in S1 File. A panel consisting of a medical microbiologist and a public health doctor were appointed to validate the questionnaire and evaluate the content validity of the questionnaire in English and Malay. The finalised questionnaire was tested in pilot study of 100 respondents selected via a convenience sample. This study adapted a double-bounded dichotomous choice (DBDC) format by using a questionnaire for the statistical efficiency suggested by previous studies [15–17]. The DBDC format of this study is summarized in Fig 1.

Respondents were provided a randomly chosen bid amount with a scenario where the hypothetical HFMD vaccine was considered safe (no danger of becoming infected from the

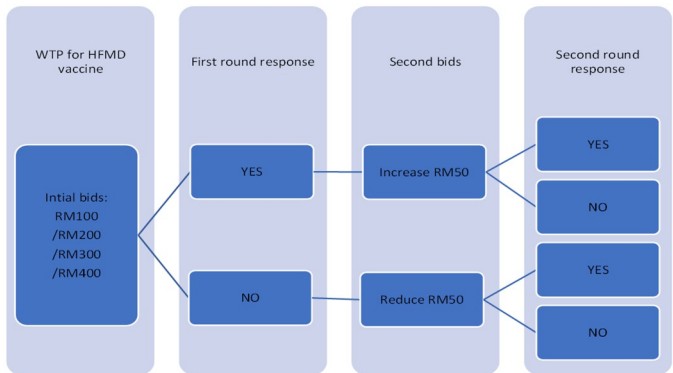

**Fig 1. Flowchart of double bounded dichotomous choice.**

vaccine), having no side-effects, comprising two doses (second dose given after one month), able to prevent HFMD infection among those who are not yet infected (but having no benefit for someone already infected) and being effective for five years. If the respondents answered "yes" to the bid amount, the amount was increased by MYR50 (equal to US$ 11.10 using a November 2022 exchange rate). However, if the respondent's response was "no" to the initial amount, the subsequent bid was decreased by MYR50 (US$11.10).

## Statistical analysis

A bivariate probit model was used to assess the key determinants of HFMD vaccine and the mean WTP was measured using the Krinsky and Robb procedure [18].

In the bivariate probit model, the WTP function takes the following form:

$$z_1^* = \delta_1\beta_1 + \varepsilon_1 \tag{1}$$

$$z_2^* = \delta_2\beta_2 + \varepsilon_2 \tag{2}$$

$z_1^*$ and $z_2^*$ is the WTP for the respondents. Both error terms $(\varepsilon_1, \varepsilon_2)$ in Eqs (1) and (2) are jointly and assumed to be normally distributed. The $\delta_1$ is a vector of the predictor variables common to both outcomes with estimators $\beta$. The $\delta_1$ and $= \delta_2$ which might affect the parents WTP for HFMD vaccination.

Observing the two binary outcomes is expected;

$$z_1 = 1 \text{ if } z_1^* > 0 \text{ and } z_1 = 0 \text{ if } z_1^* \leq 0; \tag{3}$$

$$z_2 = 1 \text{ if } z_2^* > 0 \text{ and } z_2 = 0 \text{ if } z_2^* \leq 0; \tag{4}$$

$z_1$ and $z_2$ is discrete response of the respondents for WTP questions (1 = Yes; 0 = No). According to Alberini [17], estimates of the bivariate probit model are preferred to that of the interval data logit. The correlation coefficient between the two consecutive bid error terms is close to zero. Therefore, after checking the correlation coefficients for the first and subsequent responses, the study used a bivariate probit model. To estimate a bivariate probit model, the mean of the initial question error term and the second follow up question error term are assumed independent and covariances are assumed zero [18]. Let $P^1$ be the first bid price and $P^2$ the second bid price. The WTP bounded in: $P^1 \leq WTP < P^2$ for the yes-no responses; $P^1 > WTP \geq P^2$ for the no-yes response; $WTP \geq P^2$ for a yes-yes response; and $WTP < P^2$ for

no-no response. This study assumed the WTP of the respondents between the lowest ($WTP_i^L$) and the highest value ($WTP_i^H$). In this study, the mean WTP was calculated using the Krinsky-Robb method as follows:

$$\text{Mean} = \frac{-\bar{X}\beta'}{\beta_0} \tag{5}$$

Where, $\bar{X}$ is the row vector of the sample, including 1 for the constant term. The $\beta'$ is the column vector of estimated coefficients. The $\beta_0$ is the coefficient on the bid variable. The model used effects coding for categorical variables. All the statistical analysis performed using STATA 13 and the significance was assessed using alpha 0.05.

## Results

### Participant characteristics

The data collection for this study was planned to take place from January 13 to April 1, 2020. A total of 770 self-administered questionnaires were distributed to selected kindergartens. However, due to coronavirus disease 2019 (COVID-19) outbreak, the survey was stopped on March 16, 2020. As of March 16,690 participant responses during the study period between 13 January 2020 and 16 March 2020 and 248 data were excluded from the final analysis due to missing information. A total of 442 (64.05%) participants, well-distributed from regions of the district resulted. However, 52 respondents were eliminated due not being willing to pay for HFMD and the remaining 390 were analysed. Table 2 shows the parents' profile, where 272 (70.26%) were the mother, and 116 (29.74%) were the father. Most of the parents were aged 31 to 40 (72.05%) with a small proportion aged 51 and 60 (1.79%). Most respondents were Malay (64.87%) followed by Chinese (23.85%) and Indian (10.51%). Around 41% of the parents had a bachelor's degree qualification, and 21.28% had only a high school education or below. Of note, 32.56% of the parent's household income was RM7000 and above, and only 3.33% were RM1000 and below.

The distribution of the double-bounded dichotomous choice responses shows that the higher the initial bids, the lower the percentage of YES-YES percentage and the percentage of NO-NO also increases (Table 3). For some parents, if they accept the initial bids, the higher bids will be given and vice versa. Interestingly, 10.8% of the parents rejected the price increase and 17.8% accepted reduced bids.

This study used a bivariate probit model to estimate the WTP based on the error correction between the first and second responses for the bid's questions. The correlation between first responses for the WTP question and the second WTP question had a result of 0.39 and was statistically significant at 1%. This indicates that the separate probit model will not be efficient to estimate the WTP value [18]. Furthermore, the Rho correlation coefficient being statistically significantly indicated that the bivariate probit models are appreciated [19]. A Variance Inflation Factor (VIF) was computed to check the existence of serious multicollinearity problems among continuous and discrete explanatory variables. The result indicated no multicollinearity problem exists among the explanatory variables with a mean VIF was 3.27.

The bivariate probit model was used to identify explanatory variables that influence parents' WTP. As suggested by Mitchell and Carson [20], a robust estimator was used to control the potential bias of non-normality and outliers in contingent valuation studies. This study used bivariate probit robust estimation to reduce the effects of heteroscedasticity.

The relationship between the responses for single and double bounded choices with the independent variables was checked using a chi-square test. The Chi-square test was used to test the variable having a relationship with the response's variables. The result of chi-square is

**Table 2. Description of socio-economics variables.**

| Characteristics | Frequency (N) | % |
|---|---|---|
| **Gender** | | |
| Male | 116 | 29.74 |
| Female | 274 | 70.26 |
| **Age** | | |
| 20–30 | 29 | 7.44 |
| 31–40 | 281 | 72.05 |
| 41–50 | 73 | 18.72 |
| 51–60 | 7 | 1.79 |
| **Race** | | |
| Malay | 253 | 64.87 |
| Chinese | 93 | 23.85 |
| Indian | 41 | 10.51 |
| Others | 3 | 0.77 |
| **Education** | | |
| High School or below | 83 | 21.28 |
| Certificate or Diploma | 121 | 31.03 |
| Bachelor's Degree | 158 | 40.51 |
| Postgraduate Education | 28 | 7.18 |
| **Household income** | | |
| <RM1000 | 13 | 3.33 |
| RM1001-RM3000 | 106 | 27.69 |
| RM3001-RM5000 | 101 | 25.90 |
| RM5001-RM7000 | 41 | 10.51 |
| RM7001 & above | 127 | 32.56 |

shown in Table 4. Only income and education associate with single and double bounded responses.

The chi-square test of association, income and education variables is included in the univariate probit model (Table 5). The result shows that high school or below, certificate or diploma, bachelor's degree, income ranged MYR1001 to 3000 and MYR7000 and above are significantly determine single bounded WTP. However, for double bounded, the determinants are found for parents with poor education background, high and low income.

Table 6 shows that education and income have an impact on the WTP. The coefficient value of the single bound, which is 0.001, shows that with a 1% increase in bid the probability

**Table 3. The distribution of the dichotomous responses.**

| Initial bids | Yes-Yes | Yes-No | No-Yes | No-No | Observation |
|---|---|---|---|---|---|
| RM100 (USD $22.20) | 40 (66.6%) | 3 | 12 | 6 (9.8%) | 61 |
| RM150 (USD $33.30) | 32 (52.5%) | 6 | 16 | 7 (11.5%) | 61 |
| RM200 (USD $44.40) | 32 (48.5%) | 11 | 10 | 13 (19.7%) | 66 |
| RM250 (USD $55.50) | 28 (50%) | 3 | 8 | 17 (30.4%) | 56 |
| RM300 (USD $66.60) | 17 (35.4%) | 3 | 12 | 16 (33.3%) | 48 |
| RM350 (USD $77.70) | 32 (49.2%) | 8 | 8 | 17 (26.1%) | 65 |
| RM400 (USD $88.80) | 13 (39.4%) | 8 | 3 | 9 (27.3%) | 33 |
| Total | 194 (49.7%) | 42 (10.8%) | 69 (17.8%) | 85 (21.8%) | 390 |

**Table 4. Chi-square test of association between the responses (single and double bounded) and each explanatory variable.**

| Explanatory variables | Single bounded | | | Double bounded | | |
|---|---|---|---|---|---|---|
| | Chi-square | df | P value | Chi-square | df | P value |
| Gender | 24.425 | 28 | 0.659 | 0.581 | 1 | 0.446 |
| Age | 0.524 | 1 | 0.469 | 34.224 | 28 | 0.194 |
| Education level | 26.477 | 3 | 0.000*** | 7.127 | 3 | 0.068* |
| Income | 90.220 | 64 | 0.017** | 79.381 | 64 | 0.093* |
| Ethnicity | 2.918 | 3 | 0.404 | 3.381 | 3 | 0.337 |

***Statistically significant at 1%,

**statistically significant at 5% and

*statistically significant at 10%

**Table 5. Probit univariate analysis.**

| Explanatory variables | Single bounded | | | Double bounded | | |
|---|---|---|---|---|---|---|
| | Parameter estimate | Std error | P value | Parameter estimate | Std error | P value |
| **Education** | | | | | | |
| High School or below | -0.716 | 0.158 | 0.000*** | -0.408 | 0.157 | 0.010*** |
| Certificate or Diploma | 0.311 | 0.142 | 0.028** | 0.183 | 0.144 | 0.205 |
| Bachelor's Degree | 0.348 | 0.133 | 0.009*** | 0.132 | 0.135 | 0.327 |
| Postgraduate Education | -0.385 | 0.246 | 0.119 | 0.013 | 0.256 | 0.961 |
| **Household income** | | | | | | |
| <RM1000 | -0.584 | 0.359 | 0.104 | -0.987 | 0.370 | 0.008*** |
| RM1001-RM3000 | -0.412 | 0.143 | 0.004*** | -0.201 | 0.146 | 0.168 |
| RM3001-RM5000 | -0.073 | 0.146 | 0.617 | -0.114 | 0.149 | 0.445 |
| RM5001-RM7000 | 0.306 | 0.218 | 0.160 | 0.359 | 0.229 | 0.118 |
| RM7001 & above | 0.411 | 0.144 | 0.004*** | 0.313 | 0.144 | 0.030** |

*statistically significant at 10%;

**statistically significant at 5%;

***Statistically significant at 1%

of parents WTP reduces by 0.1%. However, the double bounded function indicates that with a 1% increase in the second bid, the probability of parents WTP reduces by 0.3%. This implies that the higher the price (bid) of the vaccine, the lower the WTP by the parents for the HFMD vaccine.

Single bounded responses show that educated parents have greater WTP for HFMD vaccination. A 1% increase in parents who have a certificate or diploma and bachelor's degree increases their WTP by 72.5% and 64.8%, respectively. However, double bounded responses shows that a 1% increase in parents who lack education reduces their WTP by 43.5%. This implies that educated parents have more WTP for HFMD vaccine compared to unenlightened parents.

In single bounded responses, a 1% increase in parents with high income will increase their WTP by 35.6%. However, in double bounded responses, a 1% increase in low-income parents will reduce their WTP by 71.6%. This suggests that higher parents' income significantly improves their WTP for HFMD vaccine.

The value of rho is estimated at 0.717, and it is positive signed and statistically is significantly different from zero in a Wald test, indicating that the outcome is related and some

**Table 6. Parameter estimates for seemingly unrelated bivariate probit model.**

| Explanatory variables | Single bounded function | | | Double bounded function | | |
|---|---|---|---|---|---|---|
| | Coeff. | Robust Std error | p-value | Coeff. | Robust Std error | p-value |
| **Price** | | | | | | |
| Start bid | -0.001 | 0.001 | 0.073* | | | |
| Follow-up bid | | | | -0.003 | 0.001 | 0.000*** |
| **Education** | | | | | | |
| High School or below | 0.035 | 0.285 | 0.903 | -0.435 | 0.167 | 0.009*** |
| Certificate or Diploma | 0.725 | 0.258 | 0.005*** | | | |
| Bachelor's Degree | 0.648 | 0.237 | 0.006*** | | | |
| **Household income** | | | | | | |
| <RM1000 | | | | -0.716 | 0.282 | 0.011** |
| RM1001-RM3000 | -0.144 | 0.143 | 0.314 | | | |
| RM7001 & above | 0.356 | 0.166 | 0.032** | | | |
| Constant | -0.004 | 0.318 | 0.988 | 0.212 | 0.146 | 0.146 |
| RHO | 0.717 (0.064) | | | | | |
| Wald test rho = 0; Chi2(1) = 65.82; p-value>Chi2 = 0.000 | | | | | | |

***Statistically significant at 1%,;

**statistically significant at 5% and;

*statistically significant at 10%

**Table 7. Mean and median of Krinsky and Robb procedure.**

| | Mean = Median | 95% lower bound | 95% upper bound | ASL[a] |
|---|---|---|---|---|
| Single bounded | 460.23 | -453.52 | 1926.26 | 0.033 |
| Double bounded | 394.00 | 335.30 | 517.64 | 0.000 |

[a] ASL = Achieved Significance Level for testing H0: WTP< = 0 versus H1: WTP>0

unobserved factors are positively related to both dependent variables. The confidence intervals around the median and mean WTP are estimates based on the Krinsky and Robb [21] procedure using 10,000 draws [22, 23].

The mean WTP estimated for single bounded was MYR460.23 (USD $102.17) and for double bounded was MYR394.00 (USD $87.47) (Table 7). The WTP for single bounded was higher compared to the double bounded response and indicates the efficiency of using the double-bounded approach.

## Discussion

Hand, foot, and mouth disease (HFMD) can cause severe illness in young children and have larger economic impacts. Taking place before a vaccine has been approved for use in Malaysia, this study surveyed parents in Selangor, Malaysia to determine their WTP for HFMD vaccination. We found that the price of the vaccine, parents' education and income are all associated with their WTP for HFMD vaccination.

According to economic theory, the law of demand, as the price of goods increases the quantity demand of goods decreases. In this study, we found that as the HFMD vaccine bids increase, WTP of the parents' decreases. Currently, there is no study investigating WTP for

HFMD, however, published studies for several types of vaccines also found the same outcome such as meningococcal conjugate vaccines [24]; dengue vaccine [25, 26]; pneumococcal vaccine [27, 28]; PCV-7 and influenza vaccine [29]; cholera vaccine [30]. This indirectly indicates that parents prefer the vaccine at the lowest price to vaccinate their children with HFMD vaccination.

This study found a strong relationship between education and willingness to pay for vaccination. In particular, WTP was relatively high for those with a certificate or bachelor's degree, and relatively low with those who only had a high school education or less. In the double bounded analysis, we also see that parents with low education reduce their WTP. This finding is in line with previous research [31]. A possible explanation is that individuals of lower education may also have less knowledge of HFMD, or reduced risk perceptions [32] relative to others. A more direct explanation, however, would be that education and income are highly correlated, and those with lower education have lower disposable income and reduced ability to pay out of pocket for a HFMD vaccine.

Similar to education, household income also significantly influenced the parents' WTP on HFMD vaccine in our findings. The same outcome seen in other studies [30, 31]. Furthermore, Olson et al. [33]. highlighted those financial constraints due to having a larger family decreased WTP for vaccination. Interestingly, parents who earn high-income increase WTP for HFMD vaccination at a given price (single bounded) and low-income parents reduce their WTP if price increases (double bounded). This indicates that parental income influences their decision on vaccination based on vaccine price. We note that a previous study in China examining preferences for vaccination programs, individuals with higher incomes actually preferred higher cost vaccines, which may have been tied to perceptions about quality of the vaccine [34].

Lower WTP for vaccine among individuals with lower education and lower incomes indicates that when EV71 vaccination becomes available in Malaysia, there could be substantial socioeconomic disparities in the absence of government subsidies. Overall, the reduced WTP for EV71 vaccines among families with low education and low incomes indicates problems in a future rollout of the vaccine. Absent government subsidies, there could be substantial socioeconomic differences in vaccination uptake, which could lead to disparities in incidence of disease and the economic burden associated with disease.

The estimated mean value for HFMD vaccine in this study was RM394.00 for two doses, which is a higher value estimated for different vaccinations in Malaysia such as dengue vaccine MYR 39.21 per dose [35]; Hepatitis B vaccine RM303 for three doses [36] and coronavirus disease 2019 (COVID-19) vaccine RM134 for a single dose [37]; According to economic theory, the equilibrium of the price is determined by demand and supply of the product. This study suggests that the HFMD vaccine's price is the key determinant of parents' WTP for the HFMD vaccine if the vaccine is available in the future (demand side). Therefore, the manufacture (supply side) of the vaccine should use the cost-plus pricing setting [38], where the seller is paid a mark-up over the marginal cost and is able to produce the vaccine at the lowest cost. With that, HFMD vaccine would be available in the future at a more affordable price for a larger share of the population.

## Limitations

There are several possible limitations to this study. Convenience-based selection of the sites could have led to bias in our estimates and limited generalizability. This may have been compounded by us being unable to finish data collection due to the COVID-19 pandemic. We note that many factors may influence WTP for a vaccine. Although the education and income, as measures of lower socioeconomic status, emerged as strongly associated with WTP levels,

there could be other structural barriers and personal beliefs or attitudes, not measured in this study, that influence WTP.

## Conclusions

This study used a bivariate probit model to analyse the parents WTP on HFMD vaccination in Selangor, Malaysia. We found that 71.5% of the parents were willing to pay for HFMD vaccination. The estimated mean WTP for HFMD vaccine was RM394 (USD $87.47) for two doses. The result shows that the higher the price (bid) of the vaccine, the lower the parents' WTP for HFMD vaccine. Moreover, parents' income and education significantly increase their WTP for HFMD vaccine. Based on our finding, there is a demand for the HFMD vaccine in Malaysia if available in the private market. However, the price of the vaccine should use a price discrimination strategy such as a sliding scale where a different price is charged based on the parent's income in order to mitigate future disparities in HFMD incidence. Moreover, awareness of HFMD should increase among the low educated parents to increase their WTP for HFMD if available in the private market in the future.

## Supporting information

**S1 File. Questionnaire used in this study.**
(PDF)

## Acknowledgments

The authors would like to thank the questionnaire moderator committee from the Faculty of Health Sciences, Universiti Putra Malaysia: Dr. Rosliza Abdul Manaf, Department of Community Health, and Dr. Siti Zulaikha Binti Zakariah, Department of Medical Microbiology and Parasitology.

## Author Contributions

**Conceptualization:** Yogambigai Rajamoorthy, Harapan Harapan.

**Data curation:** Yogambigai Rajamoorthy, Niazlin Mohd Taib, Subramaniam Munusamy.

**Formal analysis:** Yogambigai Rajamoorthy.

**Funding acquisition:** Yogambigai Rajamoorthy.

**Investigation:** Yogambigai Rajamoorthy, Subramaniam Munusamy.

**Methodology:** Yogambigai Rajamoorthy, Subramaniam Munusamy.

**Project administration:** Yogambigai Rajamoorthy, Niazlin Mohd Taib, Subramaniam Munusamy.

**Supervision:** Abram Luther Wagner.

**Validation:** Harapan Harapan, Abram Luther Wagner.

**Writing – original draft:** Yogambigai Rajamoorthy, Niazlin Mohd Taib.

**Writing – review & editing:** Niazlin Mohd Taib, Harapan Harapan, Abram Luther Wagner.

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
