## [Decision Letter · Decision Letter 0]

16 Mar 2023

PONE-D-22-33005Application of the Double Bounded Dichotomous Choice Model to the Estimation of Parent’s Willingness to pay for the Hand Foot Mouth Disease Vaccination: A survey in Selangor, MalaysiaPLOS ONE

Dear Dr. Rajamoorthy,

Thank you for submitting your manuscript to PLOS ONE. After careful consideration, we feel that it has merit but does not fully meet PLOS ONE’s publication criteria as it currently stands. Therefore, we invite you to submit a revised version of the manuscript that addresses the points raised during the review process. Dear Authors,  Please be specific and brief in your response to reviewers' queries.

We look forward to receiving your revised manuscript.

Kind regards,

Muhammad Khalid Bashir, PhD

Academic Editor

PLOS ONE

Journal Requirements:

Reviewers' comments:

Reviewer's Responses to Questions

**Comments to the Author**

1. Is the manuscript technically sound, and do the data support the conclusions?

Reviewer #1: Yes

Reviewer #2: Yes

2. Has the statistical analysis been performed appropriately and rigorously? 

Reviewer #1: Yes

Reviewer #2: Yes

3. Have the authors made all data underlying the findings in their manuscript fully available?

Reviewer #1: Yes

Reviewer #2: Yes

4. Is the manuscript presented in an intelligible fashion and written in standard English?

Reviewer #1: Yes

Reviewer #2: No

5. Review Comments to the Author

Reviewer #1: The article "Application of the Double Bounded Dichotomous Choice Model to the Estimation of Parent’s Willingness to pay for

the Hand Foot Mouth Disease Vaccination: A survey in Selangor, Malaysia" scientifically well written and hope will caught the attention of readership and stakeholders as well. Some of my comments and suggestion are given below:

1. The Introduction part is very small, authors should improve this section with more citations and they can increase the review of literature to support their study significantly.

2. I think authors should provide a table of samples which clearly indicating the name of district, number of kindergartens schools visited and respondents from each district then finally total of them (in Material and Methods Section).

3. Here the dependent variables Z1* and Z2* not explained and also list the names of predictor variables in equation 1 and 2

Either the same variables used in both equations or different in each equation? (from empirical model part).

4. I think authors should include some more explanatory variables in model like family origin (rural or urban) and family size as they can considerably influence the WTP and hope that these questions were also asked from the respondents during the survey. With the inclusion of such variables in the model these results would be more reliable and model will be suitable for future predictions.

Finally the study is good and most probably it will considerable contribution in the chunk of literature.

Reviewer #2: I have reviewed the article entitled “Application of the Double Bounded Dichotomous Choice Model to the Estimation of Parent’s Willingness to pay for the Hand Foot Mouth Disease Vaccination: A survey in Selangor, Malaysia”. I appreciate the effort of the authors but I have some questions and suggestion to increase the manuscript worth.

The introduction part must be extended? Author should explain why Govt. has not approved the vaccine yet? And what is the alternate? And how the vaccine is available in country for which the WTP of parents is to be measured?

Line 83-87: it could be better to insert the formula that have been used by the authors to have 384 respondents as sample size. moreover, the selection criteria must be briefly incorporated. Moreover, the author should explain that what they have followed that criteria fully or partially. How much has the value of P used in formula?

Moreover, the author did not explain how they collected the data? During COVID time?

Line156-158: I could not understand that how the number of respondents has fallen from 770 to 690?

The authors must incorporate the literature review section?, and should explain the research gap that they want to fill?

Authors should incorporate the limitations of the study?

Best of Luck

6. PLOS authors have the option to publish the peer review history of their article (what does this mean?). If published, this will include your full peer review and any attached files.

Reviewer #1: **Yes: **Dr. Qasir Abbas (Institute of Agricultural & Resource Economics, University of Agriculture, Faisalabad, Pakistan)

Reviewer #2: No

---

## [Author Response · Author response to Decision Letter 0]

3 May 2023

Reviewer 1#:

1. The Introduction part is very small, authors should improve this section with more citations and they can increase the review of literature to support their study significantly.

Responses: 

Thank you for the comment. We have added in several references.

Hand foot and mouth disease (HFMD) is a disease caused by several viruses within the Picornaviridae family, notably enterovirus 71 (EV71) and coxsackievirus A16 [1,2].

HFMD occurs worldwide, and its global emergence necessitates enhanced surveillance of diseases [3,4]. Within Malaysia, the outbreaks of HFMD cases are typically due to EV71[3,5]. 

Beyond the direct health effects of an EV71 infection, HFMD outbreaks could result in larger socioeconomics consequences [7]. Caring for children with HFMD results in parental absenteeism from work [8]. Moreover, large HFMD outbreaks could result in closure of childcare centres compounding these issues [8].

Since 2015, three inactivated EV-71 vaccines have been approved in China [7] and effectively reduce the severe HFMD cases and mortality caused by EV71[10].

2.I think authors should provide a table of samples which clearly indicating the name of district, number of kindergartens schools visited and respondents from each district then finally total of them (in Material and Methods Section).

Responses: 

The table of sample size calculated is included. The kindergarten selection published elsewhere citation included. However, the selection method discussed. To represent the population, several kindergartens from each district were randomly selected. The sample size and respondents’ selection criteria were a part of a Malaysian HFMD Project that have been described elsewhere [12,13].

The participants were selected using a three-stage cluster sampling method. Using the nine administrative districts in Selangor as a sampling frame, the number of samples from each district was calculated based on its population size proportion (i.e. high numbers in some districts and low in some districts). In the second stage, a convenience sample of kindergartens was selected based on kindergarten lists in each district. In the last stage, a convenience sample of participants was recruited from each kindergarten using a quota to meet each district’s calculated sample size. Before starting the study, we doubled the sample size to 768 (≈770) to avoid insufficient sample size due to incomplete data.

3. Here the dependent variables Z1* and Z2* not explained and also list the names of predictor variables in equation 1 and 2

Either the same variables used in both equations or different in each equation? (from empirical model part).

Responses: 

The explanation for Z1* and Z2* included.

The same alphabet used to explain the Willingness to pay responses. 

4. I think authors should include some more explanatory variables in model like family origin (rural or urban) and family size as they can considerably influence the WTP and hope that these questions were also asked from the respondents during the survey. With the inclusion of such variables in the model these results would be more reliable and model will be suitable for future predictions.

Responses: 

Thank you for the suggestion. We will consider the suggested variables for future studies. 

Reviewer 2#:

1. The introduction part must be extended?

Responses: 

Please see our response to the previous reviewer.

2. explain why Govt. has not approved the vaccine yet? And what is the alternate? And how the vaccine is available in country for which the WTP of parents is to be measured?

Responses: 

Recent development of an EV71 vaccine has emerged as an important tool to control HFMD [4,9]. Treatment of HFMD is largely supportive with no specific therapies available [2]. Since 2015, three inactivated EV-71 vaccines have been approved in China [7] and effectively reduce the severe HFMD cases and mortality caused by EV71[10]. However, the vaccines have not yet been approved in Malaysia leaving the country with no effective prevention or treatment methods for HFMD [11]. 

In the future, the EV71 vaccine may be approved for use within Malaysia. However, it is yet unclear how well accepted the vaccine will be by the general population. Presumably, parental willingness to accept the EV-71 vaccines will be affected by its efficacy, side effects, and cost. There are only two studies published in China on the willingness of parents to vaccinate [5,7]. However, the willingness to pay (WTP) for HFMD vaccination has yet to be estimated and could also differ between countries based on their epidemiological background and population experiences with disease. Therefore, this study’s main objective was to estimate WTP of parents for HFMD and identify its determinants by employing the contingent valuation method in Selangor state.

3. Line 83-87: it could be better to insert the formula that have been used by the authors to have 384 respondents as sample size. moreover, the selection criteria must be briefly incorporated. Moreover, the author should explain that what they have followed that criteria fully or partially. How much has the value of P used in formula?

Responses: 

The sample size calculation included 

4. Moreover, the author did not explain how they collected the data? During COVID time?

Responses: 

The data collection for this study was planned to take place from January 13 to April 1, 2020. A total of 770 self-administered questionnaires were distributed to selected kindergartens. However, due to coronavirus disease 2019 (COVID-19) outbreak, the survey was stopped on March 16, 2020.

5. Line156-158: I could not understand that how the number of respondents has fallen from 770 to 690?

Responses: 

The respondents number fell due to the COVID-19 outbreak. The actual data collection estimated 770, however, before the movement control the actual data collected was 690. 

A total of 770 self-administered questionnaires were distributed to selected kindergartens. As of March 16, 690 participant responses during the study period between 13 January 2020 and 16 March 2020 and 248 data were excluded from the final analysis due to missing information.

6. The authors must incorporate the literature review section?, and should explain the research gap that they want to fill?

Responses: 

Recent development of an EV71 vaccine has emerged as an important tool to control HFMD [4,9]. Treatment of HFMD is largely supportive with no specific therapies available [2]. Since 2015, three inactivated EV-71 vaccines have been approved in China [7] and effectively reduce the severe HFMD cases and mortality caused by EV71[10]. However, the vaccines have not yet been approved in Malaysia leaving the country with no effective prevention or treatment methods for HFMD [11].

A possible explanation is that individuals of lower education may also have less knowledge of HFMD, or reduced risk perceptions [32] relative to others. A more direct explanation, however, would be that education and income are highly correlated, and those with lower education have lower disposable income and reduced ability to pay out of pocket for a HFMD vaccine.

We note that previous study in China examining preferences for vaccination programs, individuals with higher incomes actually preferred higher cost vaccines, which may have been tied to perceptions about quality of the vaccine[34]. 

7. Authors should incorporate the limitations of the study?

There are several possible limitations to this study. Convenience-based selection of the sites could have led to bias in our estimates and limited generalizability. This may have been compounded by us being unable to finish data collection due to the COVID-19 pandemic. We note that many factors may influence WTP for a vaccine. Although the education and income, as measures of lower socioeconomic status, emerged as strongly associated with WTP levels, there could be other structural barriers and personal beliefs or attitudes, not measured in this study, that influence WTP.

---

## [Decision Letter · Decision Letter 1]

26 May 2023

Application of the Double Bounded Dichotomous Choice Model to the Estimation of Parent’s Willingness to pay for the Hand Foot Mouth Disease Vaccination: A survey in Selangor, Malaysia

PONE-D-22-33005R1

Dear Dr. Rajamoorthy,

We’re pleased to inform you that your manuscript has been judged scientifically suitable for publication and will be formally accepted for publication once it meets all outstanding technical requirements.

Kind regards,

Muhammad Khalid Bashir, PhD

Academic Editor

PLOS ONE

Additional Editor Comments (optional):

Authors to please carefully incorporate reviewer 2's suggestions in revision

Reviewers' comments:

Reviewer's Responses to Questions

**Comments to the Author**

1. If the authors have adequately addressed your comments raised in a previous round of review and you feel that this manuscript is now acceptable for publication, you may indicate that here to bypass the “Comments to the Author” section, enter your conflict of interest statement in the “Confidential to Editor” section, and submit your "Accept" recommendation.

Reviewer #1: All comments have been addressed

Reviewer #2: All comments have been addressed

2. Is the manuscript technically sound, and do the data support the conclusions?

Reviewer #1: Yes

Reviewer #2: Yes

3. Has the statistical analysis been performed appropriately and rigorously? 

Reviewer #1: Yes

Reviewer #2: Yes

4. Have the authors made all data underlying the findings in their manuscript fully available?

Reviewer #1: Yes

Reviewer #2: Yes

5. Is the manuscript presented in an intelligible fashion and written in standard English?

Reviewer #1: Yes

Reviewer #2: Yes

6. Review Comments to the Author

Reviewer #1: the authors have responded all the queries that were raised in the first round of review. I am satisficed with the revised version of manuscript and suggested that to accept the manuscript for publication.

Reviewer #2: The authors have incorporated just a few lines in the introduction. I suggest that they also incorporate the description of why they feel the need to measure the willingness of parents to receive vaccines. Is there any problem that in this country the people are not willing to pay for vaccines? Authors should clearly create the link between these two concepts: vaccine and willingness, and then they must state why they have focused on this.

Lines 129–130: authors must combine these lines. They should avoid single-line paragraphs.

7. PLOS authors have the option to publish the peer review history of their article (what does this mean?). If published, this will include your full peer review and any attached files.

Reviewer #1: **Yes: **Dr. Qasir Abbas

IARE, UAF, Pakistan.

Reviewer #2: No

---

## [Editor Report · Acceptance letter]

2 Jun 2023

PONE-D-22-33005R1 

Application of the Double Bounded Dichotomous Choice Model to the Estimation of Parent’s Willingness to pay for the Hand Foot Mouth Disease Vaccination: A survey in Selangor, Malaysia 

Dear Dr. Rajamoorthy:

I'm pleased to inform you that your manuscript has been deemed suitable for publication in PLOS ONE. Congratulations! Your manuscript is now with our production department. 

Kind regards, 

on behalf of

Dr. Muhammad Khalid Bashir 

Academic Editor

PLOS ONE